# Functional Chitosan–Calcium Carbonate Coatings for Enhancing Water and Fungal Resistance of Paper Materials

**DOI:** 10.3390/molecules27248886

**Published:** 2022-12-14

**Authors:** Erwan Le Goué, Frédérique Ham-Pichavant, Stéphane Grelier, Jordan Remy, Véronique Coma

**Affiliations:** 1Laboratoire de Chimie des Polymères Organiques, Université de Bordeaux, CNRS, Bordeaux INP, UMR 5629, 16 Avenue Pey-Berland, 33600 Pessac, France; 2Papeterie Zuber Rieder, rue Ernest Zuber, 25320 Boussières, France

**Keywords:** hydrophobicity, antifungal activity, stearic acid, cellulose, *Chaetomium globosum*

## Abstract

The objective of this study was to increase the water resistance of paper while providing fungal resistance using a bio-based coating made from chitosan. The water resistance was improved through the surface control of roughness using modified calcium carbonate particles. The higher the quantity of particles in the film-forming solution, the higher the surface hydrophobicity of the paper. The addition of particles was found to counterbalance the chitosan hydrophilicity through the control of the coatings’ penetration in the paper bulk. As a consequence, the wetting time and liquid water resistance were enhanced. The antifungal activity of the film-forming solutions and coated paper was also investigated against the growth of *Chaetomium globosum*, which was selected as a model strain able to contaminate paper materials. The results reveal that the antifungal activity of chitosan was improved by a possible synergic effect with the bicarbonate ions from the mineral particles.

## 1. Introduction

Paper, a bio-based material composed of a network of cellulose fibers, is a versatile material due to inherent characteristics such as its porous structure, and its lightweight, easily processable and high mechanical properties. Paper is attractive for various applications such as packaging [1,2,3], electronics [4], sensors [5] and microfluidic substrates [6], but its use in high moisture conditions or long storage is limited due to its water and fungal sensitivity.

To improve the water resistance of paper-based materials, surface modification using coatings is commonly performed, and numerous attractive studies have investigated hydrophobic coatings containing particles and binder associations [7,8,9]. Indeed, particles can develop a surface roughness, which reduces the contact between the water and surface material [10]. This property is the basic concept of the Lotus effect, notably due to micro- and nanostructured water-repellent surfaces leading to rolling-off water drops.

Among the common mineral particles used for surface coatings, precipitated calcium carbonate (PCC) is an affordable filler with a scalenohedral structure used in the paper industry. The modification of PCC with fatty acid enhanced the hydrophobic character and, thus, the material’s water resistance [11,12]. As an example, the water resistance of hydrophobic coatings is obtained thanks to a one-pot waterborne coating with various amounts of modified PCC and petroleum-based latex [13].

Similar results were also described using modified cellulose nanofibers as binders. A two-step process was developed to increase surface roughness and hydrophobicity using a coating with PCC and cellulose nanofibers (CNF), followed by a dip-coating treatment in an alkyl ketene dimer solution [14]. More recently, a one-step superhydrophobic coating was developed through the association of hydrophobic cellulose nanofibers with modified aragonite particles [15].

Chitosan is also used for paper coatings to improve the water barrier properties [1,2,16,17,18]. Chitosan is a biopolymer obtained through the alkaline deacetylation of chitin, whose structure is composed of glucosamine and *N*-acetylated glucosamine units. However, chitosan provides limited liquid water barrier properties as demonstrated by Bordenave [1]. Chitosan-coated paper exhibited a decrease in water penetration but an increase in surface hydrophilicity. Thus, the association of modified PCC with chitosan might counterbalance the surface hydrophilicity of chitosan-coated paper. In addition, few studies were reported on the development of bio-based coatings with low water sensitivity and bioactive properties.

Chitosan salts can confer microbial resistance to paper against several strains, especially against fungal growth [19]. Chitosan was also reported to be efficient against spore germination and germ tube elongation [20]. In addition, chitosan coatings were used to exhibit antibacterial properties [21,22]. Chitosan derivatives were also synthetized and tested as antimicrobial agents against various strains. *Penicillium natatum* was, for example, completely inhibited by alkyl–chitosan in comparison with quaternary and carboxymethyl chitosan [23]. Regarding the mode of action, it is well-known that bioactive properties are mainly obtained in acidic medium when the amino groups of chitosan are protonated [24], but the antimicrobial mechanisms are not completely understood [20].

In this work, to improve the paper properties while maintaining the potential biodegradability/compostability, chitosan was used for its antifungal activity and good film-forming properties in combination with modified PCC particles bringing surface roughness and hydrophobicity to decrease the surface water interaction. Coated paper was then characterized in terms of water resistance using different methods. Finally, the impact of the coatings in terms of their antifungal bioactivity and mechanical resistance was investigated and discussed.

## 2. Results and Discussion

### 2.1. Characterization of PCC Particles

To decrease the hydrophilic character of the chitosan-based coating, modified calcium carbonate particles with stearic acid were prepared before incorporation into the chitosan film-forming solution.

Their characterization using granulometry analysis (Figure 1) showed a change in their size distribution after modification. The granulometric volume curves showed a multimodal distribution with an increase in particle size. The aggregation phenomenon occurred during the modification process. However, aggregation was limited, as shown by the monomodal distribution in the granulometric number curves.

The chemical characterization of the particles using FTIR (Figure 2) showed new bands compared to the nonmodified particles, corresponding to the asymmetric and symmetric elongation of the C-H bond (2920 and 2850 cm^−1^). The substracted spectra clearly showed C-O stretching vibrations from the carboxylate group (1583 cm^−1^) of the fatty acid. Moreover, stearic acid was completely adsorbed at the surface of the PCC on the carboxylate structure, as proven by the absence of the carbonyl band at 1704 cm^−1^ [12,25,26,27].

The thermal analysis (TGA) of the PCC particles was then carried out to complete the FTIR analysis and determine how stearic acid was deposited on the mineral surface.

The results before and after modification are showed Figure 3, and three main weight losses were obtained. The first one between 25 and 150 °C corresponded to water loss. For the temperature range between 180 and 480 °C, the degradation of the adsorbed stearic acid and hydroxyl groups at the surface of the PCC was observed. At a temperature higher than 480 °C, the weight loss was attributed to calcium carbonate decarboxylation, leading to calcium oxide [28].

The main difference when the PCC particles were modified was observed between 180 and 480 °C, and the mass loss was divided into two temperature ranges (Table 1). Concerning the PCC particles before modification, a small degradation was observed related to bounded water and the hydroxyl groups’ removal [29]. For the modified PCC, the main weight loss was observed between 180 and 380 °C, and it was attributed to the degradation of stearic acid which was physically linked [30,31]. For the second temperature range, the maximum of the degradation occurred around 420 °C and was assigned to monolayer decomposition, composed of stearic acid chemically bonded on the form of the stearate ion [29,30,31].

The TGA analysis showed that the modification of PCC was led mainly by the stearic acid physically bonded at the calcite surface, as illustrated in Figure 4. The organization of molecules at the calcite surface is dependent on the stearic acid concentration when water is used as the solvent. For high stearic acid concentration up to the critical micelle concentration of stearic acid (7 × 10^−4^ mol.L^−1^ at 60 °C), there is a formation of micelles when the pH is basic. The presence of calcium ions from calcite in the medium leads to the decomposition of micelles and the precipitation of calcium stearate on the particles’ surface. Moreover, hydrophobic interactions between the organic tails of stearate may lead to the presence of multilayers [31].

The investigation of the hydrophobicity of the particles before and after modification was also studied using water contact angle (WCA) measurements. Without any modification, the direct absorption of water drops was observed. In contrast, a WCA of 121 ± 2° was measured after modification, indicating an increase in hydrophobicity.

### 2.2. Characterization of Chitosan-Based Films Loaded with PCC Particles

To study the chitosan-based coating, before deposition on paper material, modified PCCs were introduced in different amounts into a chitosan film-forming solution to prepare the composite films after film casting (Figure 5). The visual aspect shows that the higher modified PCC content resulted in the higher opacity, whiteness and surface rugosity of the composite films. In addition, up to 40 wt.%, the films lost their plasticity and became brittle.

The water contact angle (WCA) measurements on the composite films showed a higher surface hydrophobicity, except for 80 wt.% modified PCC, in comparison with the pure chitosan film (Table 2). The higher the WCA, the higher the surface hydrophobicity. However, with 80 wt.% of modified PCC, the water drop was directly absorbed. In this case, the roughness at the microscale was not sufficient to keep enough air between the film and the water drop. A change in the wetting regime from the Cassie–Baxter to Wenzel may happen [32]. In the Wenzel regime, the surface of the solid is completely wet, leading to higher hydrophilicity for the hydrophilic surface [33]. For the lower amount of modified PCC, after a few seconds, the film in contact with water swelled. As a consequence, it is assumed that the modified PCC decreased the hydrophilic behavior of the chitosan composite films, but the films could not be turned hydrophobic.

### 2.3. Characterization of Coated Paper

Taking into account the previous observations, paper was coated with selected formulations containing 20 and 50 wt.% of modified PCC. Table 3 summarizes the thickness and grammage of the coatings and paper. As already observed in the literature, the coating showed no impact on the paper thickness due to the lower coated grammage and coating penetration in the paper [17].

The coating penetration was investigated using FTIR-ATR characterization. As expected, the infrared spectra of the coated paper on both sides showed cellulosic absorption bands (Figure 6) at 1630 cm^−1^ (O-H bending vibration from absorbed water), 1420 cm^−1^ (C-H deformation), 1370 cm^−1^ (C-H bending), 1330 cm^−1^ (CH_2_ wagging), 1155 cm^−1^ (asymmetric deformation of C-O-C), 1110 cm^−1^ (C_2_-O wagging) and 895 cm^−1^ (C-H deformation of anomeric carbon) [34,35,36].

On the coated side, the characteristic bands of chitosan were found at 1640 cm^−1^ (C=O stretching of amide I) and 1583 cm^−1^ (*N*-H bond), in addition to the cellulose ones. Moreover, the intensity of the cellulose band at 1110 cm^−1^ decreased after the coating, suggesting a partial coverage of the cellulose fibers by chitosan. The band at 873 cm^−1^ corresponding to CaCO_3_ [37] has a higher intensity of formulation containing more calcium carbonate particles.

Considering the FTIR spectra of the coatings containing 50 wt.% of mineral particles, the intensity of the characteristic bands of chitosan and CaCO_3_ was more pronounced, suggesting a better retention of the coatings on the paper surface due to the incorporation of the particles.

For all materials, the noncoated side showed the same FTIR spectra without new absorption bands or band-intensity modification, which indicated that the coatings did not cross through the paper.

The morphology of the coated papers was studied using SEM analyses, and the views are presented in Figure 7. The paper with only the chitosan coating or 20 wt.% of modified PCC showed a lot of fibers and pores that were not completely covered by the coating in opposition to the papers coated with more modified PCC. These results were in accordance with FTIR results showing better retention of coatings at the surface of the paper when the amount of modified PCC was at a sufficient concentration in the film-forming solution. The cross-section views show the penetration of chitosan in the material, leading to a more compact bulk structure.

### 2.4. Liquid Water Resistance

The investigation of liquid water resistance was first carried out using the measurements of WCA, surface wettability, wetting time and Cobb_60_ (Table 3). The WCA provided information about the surface hydrophobicity. The high WCA (110°) exhibited by the uncoated paper is due to the sizing used for its manufacture. After a coating with pure chitosan, the contact angle decreased to 82°, leading to the lowest WCA. As expected, chitosan provided the hydrophilicity of the paper. The introduction of modified PCC in the formulation reduced the hydrophilicity due to the chitosan developing a higher surface roughness. The highest WCA was obtained with the formulation containing 50 wt.% of modified PCC.

The PDA curves inform us about how the liquid water wets the paper. The surface wettability (W), wetting time and water penetration profile are shown in Table 3 and Figure 8, respectively. The curves showed two regions. The first one corresponded to the paper surface coming into contact with liquid water. The paper surface became wet when air was evacuated (the maximum of the curve). The second region was related to the rate of water penetration in the bulk material (the curve slope). The PDA curves showed similar behavior between coatings and allowed us to calculate the surface wettability and wetting time. The surface wettability (W) was assessed using the measure of the area between the ordinate axis and PDA curve at the wetting time. The higher the value, the lower the surface wettability [38]. The highest value was found for the coating with 50 wt.% of modified PCC. The results show that surface wettability was not only dependent on the surface hydrophobicity provided using the WCA measurements. The dynamic of water penetration was related to the evacuation of the air layer at the surface of the paper. The combination of higher rugosity and higher WCA led to a lower surface wettability.

The wetting time was correlated to the surface roughness of coatings, i.e., the higher roughness, the higher the wetting time. Coating with 50 wt.% of modified PCC increased the wetting time of the paper by 172%. Considering previous characterizations, adding modified mineral particles with chitosan improved the coatings’ retention at the surface of paper and allowed the counter-balance with the chitosan hydrophilicity.

The Cobb_60_ value corresponds to the amount of liquid water absorbed by paper after 60 s of contact. The smaller the value, the higher the liquid water barrier properties. The coatings reduced the water absorptiveness (15%). However, the reported values were similar for all coatings, including the chitosan coating. The improvement in the surface hydrophobicity by the coating was not sufficient to drastically decrease the liquid water absorption. The best liquid water barrier was obtained with the 50 wt.% coating formulation. Clearly, the coatings with minerals had more impact on the surface properties than the bulk properties.

Water resistance is also illustrated by the water vapor permeability of materials. The results of the WVTR for the coated papers are shown in Table 4. The chitosan coating led to a reduction in the water vapor transmission rate (WVTR) by 18% in our conditions, which was higher than the coated paper with particles. As previously observed using the SEM micrographs, the modified PCC maintained coatings at the surface of the paper. In that case, the lowest chitosan penetration led to a decrease in the efficiency of the coating to fill preferential water ways. As a result, the water vapor permeability appeared to be more highly affected by the bulk properties than the surface properties.

According to the literature, the influence of chitosan on the water barrier properties of coated paper is still under debate. Some authors observed that chitosan did not modify the WVTR [17,39], which is in contradiction with Vartiainen’s results and other publications [1,40,41], which showed a significant reduction in WVTR [22]. This could be due to the experimental conditions, such as the relative humidity adjusted for the tests, but also to the chitosan properties (molar mass and deacetylation degree).

Finally, as previously shown on the chitosan films or coatings, NMR relaxometry is a useful method to investigate the water interaction with a material [1,42]. The spin–spin relaxation time (*T*_2_) is related to the fast interactions between water molecules and their macromolecular environment showing two types of populations called free or bounded water in the cellulose fiber network (Table 5). The investigation of water–cellulose interactions provides information about paper hydrophobicity.

According to the literature, the lowest relaxation time, *T*_2*a*_, for uncoated and coated papers was assigned to the first layers of bounded water on the polymer. Relaxation time, *T*_2*b*_, is related to water-bounded molecules trapped in cell lumen [43]. The *T*_2*b*_ values were similar between all paper samples, which meant that the surface of the lumen fiber is not modified by chitosan treatment.

The relaxation times associated with the chitosan film showed a main population at *T*_2*a*,_ representing 95% of the signal intensity and was extremely tight to the polymer backbone, as discussed in [42]. In addition, the chitosan film was very hydrophilic and contained 22.5% of water. Despite low coated grammage, the paper coated with chitosan increased its initial water content from 4.3 to around 5.0% for coatings with modified PCC compared with 7.2% for the only chitosan coating. The increase in the water content was related to the coating penetration in the paper and the surface fiber coverage by chitosan. For all materials, *T*_2*a*_ was lower after coating, showing a reduction in the water molecule mobility, meaning an increase in fiber hydrophilicity [1]. This behavior could be caused by strong interactions between the cellulose fibers and chitosan.

The relaxometry data led to a better comprehension of the previous water resistance results. Generally, to control the paper water absorption, two ways were chosen: cellulose hydrophobic modification and the reduction of water’s physical access to the cellulose fibers. In the case of chitosan-based coatings, the limitation of water access to cellulose was the main factor. Water absorption was mainly controlled by chitosan hydrogel formation in contact with water, despite the increasing hydrophilicity of the fibers after the chitosan coating.

### 2.5. Mechanical Properties

As can be seen in Table 6, the tensile strength was improved by the coatings in accordance with our previous results showing strong interactions between chitosan and the cellulose matrix. The pure chitosan coating led to an increase in tensile strength by 54%. The presence of modified PCC obtained the best tensile strength improvement, which was close to 80%. Unexpectedly, the elongation at the break remained unchanged by the coatings. Indeed, a plasticizing effect with the chitosan coatings was observed in the literature, which contradicts the results of the current study [1,22,40]. This could be due to the fact that chitosan with a lower molecular weight was employed, which can contribute to a reduction in such a plasticizing effect [44].

### 2.6. Antifungal Activity

The antifungal activity of the chitosan film-forming solutions containing modified PCC was evaluated against *C. globosum* by measuring the radial growth every day for 14 days. As displayed in Table 7, three series were identified as the function of the lag phase for the chitosan-coating formulation. The first one corresponds to the control and the film-forming solution containing 50 wt.% of modified PCC, and no antifungal properties were observed. In the second series, an increase in the lag phase was observed for the chitosan film-forming solution. The incorporation of modified PCC into the film-forming solution led to an increase in pH from 4.2 to close to 6.2. At this pH, there was around 67% of the ammonium groups compared with 100% at the initial pH (pKa of chitosan is close to 6.5 [45]). As a consequence, an investigation into the impact of pH on the fungal radial growth was carried out on the chitosan film-forming solution without modified PCC, but no significant difference was observed, as shown in Table 7. The nonbioactivity of the formulation with 50 wt.% of modified PCC can be explained by a large amount of chitosan aggregation on the surface of the PCC particles resulting in a reduction in chitosan availability. Finally, the last series, with the longest lag phase, corresponds to the formulation with 20 wt.% of modified PCC.

The bioactivity of the modified PCC was also investigated using HPMC in place of chitosan to evaluate the impact of the oxygen barrier on fungal growth. Indeed, HMPC, a cellulose derivative with film-forming properties, showed similar oxygen barrier properties to chitosan but without any bioactive properties [46]. The activation effect was observed with HMPC and attributed to the cellulose derivative metabolization by microbial strains, used as a carbon source (Table 7). However, the growth rate was not affected.

Considering these results and the literature [47], the association of chitosan with bicarbonate ions, from partial particle solubilization in acidic medium, may have a synergistic effect on the fungal lag phase. Chitosan is known to affect mycelial growth and the sporulation of fungi by notably acting on membrane permeability [48]. Carbonate salts caused a reduction in the cellular turgor pressure, resulting in hyphae collapse and sporulation inhibition [47]. Concerning HPMC solutions, the bicarbonate ions could present difficulties when going through the cellular membrane, reducing their bioactivity. Hence, their presence in the chitosan solution improved bioactivity due to the membrane permeability reduction caused by chitosan.

The antifungal properties of the coated paper were then determined (Table 8). A comparison with the control paper showed no delay on the lag phase and sporulation. The growth rate was mainly affected by the coatings, probably caused by a lack of the fibers’ access to the spores and nutrient medium. The chitosan coating and 20 wt.% chitosan-modified PCC coating showed no efficient antifungal effect which is attributed to the low dry matter coated on the paper. The 50 wt.% chitosan-modified PCC formulation exhibited a reduction in the radial growth rate, which could be due to the highest water surface resistance, limiting the access of the spores to the fibers and nutrient medium.

To develop higher bioactivity with the 20 wt.% chitosan-modified PCC formulation, a second coated layer was applied on the paper. No delay on the lag phase was observed, but an inhibition by 44% was obtained at day 3, and the growth rate was lower. These results showed a cumulative effect between the reduction in the surface hydrophilicity and the antifungal behavior developed by the coatings.

## 3. Materials and Methods

### 3.1. Materials

Chitosan (low molecular weight, 20–300 cP, 1 wt.% in 1 vol.% acetic acid, 75–85% deacetylated) and stearic acid (reagent grade 95%) were purchased from Sigma Aldrich (Saint-Quentin Fallavier, France). Precipitated calcium carbonate (PCC, 99%+ for analysis) and Tween 80 were obtained from Acros Organic (Fisher Scientific France, Illkirch, France). Hydroxypropylmethylcellulose, >98%, was provided by Culminal. Lactic acid, 99%, was purchased from Prolabo. Potato dextrose agar was obtained from Biokar Diagnostics (Beauvais, France).

Paper sheets were furnished by Papeterie Zuber Rieder (Boussières, France). They were prepared from a mixture of hardwood (75%) and softwood (25%) fibers. They had a basis weight of 93 g.m^−2^ and a thickness of 122 µm.

### 3.2. Elaboration of Chitosan Films and Modified PCC

PCC particles were modified according to the procedure described by Hu [11]. Briefly, PCC (20 g) was mixed with stearic acid (0.619 g) in distilled water (45 mL). Then, the mixture was heated at 75 °C for 30 min and cooled down to room temperature.

Chitosan (1 g) was solubilized overnight with acidic water (100 mL, 0.5 vol.% lactic acid) under magnetic agitation. Afterwards, the PCC suspension was added to the chitosan solution to give a series of formulations containing 0, 5, 10, 15, 20, 40, 60 and 80 wt.% of dispersed PCC. To ensure good particle dispersion, solutions were homogenized with UltraTurrax (IKA T10 Basic, OKA-Werke GmbH, Staufen, Germany) for 2 min and then degassed in an ultrasonic bath. After casting 20 mL of solution in Petri dish (diameter of 5.5 cm), chitosan films were dried in oven (40 °C) for 48 h and conditioned in climatic chamber (23 °C, 50% RH) for at least three days.

### 3.3. Elaboration of Chitosan-Coated Papers

Paper sheets were coated with the chitosan film-forming solution, as mentioned above, using a 100 µm blade at the speed of 10 mm.s^−1^. Coated papers were dried at 40 °C and stored in same conditions as the chitosan films.

### 3.4. Characterization of the Samples

#### 3.4.1. Thickness

Paper thickness was measured using a micrometer (MI-20, Adamel Lhomargy, Roissy-en-Brie, France). Measurements were taken at five random positions.

#### 3.4.2. Bendtsen Rugosity

Bendtsen rugosity was determined using a Bendtsen tester, model 3500 from Paper Testing Association (Singapore, Malaysia). Measurements were taken at five random positions.

#### 3.4.3. Infrared Spectroscopy

Infrared absorption spectra of calcium carbonate, modified calcium carbonate particles and stearic acid were obtained using the potassium bromide technique (KBr) with an FTIR spectrometer at the absorbance mode from 4000 to 400 cm^−1^ with automatic signal gain (32 scans) at a resolution of 4 cm^−1^ (PIKE technologies, Bruker, France). A KBr pellet was used as a background.

ATR-FTIR spectra of the papers were recorded on the same FTIR device using same parameters and a Gladi Atrvertex 70 module. Baseline of spectra was adjusted with the same correction (polynomial with 1 iteration and 6 points at 3800, 2280, 1900, 1262, 800 and 411 cm^−1^) and were normalized to the C-H deformation of anomeric bond at 895 cm^−1^, not affected by coating modification.

#### 3.4.4. Granulometric Analyses

Granulometric measurements were performed using a Malvern Mastersizer 2000 (Malvern Panalytical, Worcestershire, UK) with the diffraction of a He/Ne laser beam (λ = 633 nm) by the calcium carbonate particles in a water flow between two parallel glass slides.

#### 3.4.5. Thermogravimetric Analyses

Thermogravimetric analyses of PCC particles before and after modification were performed with a TGA-Q500 system from TA Instruments (New Castle, DE, USA). About 10 mg of sample was weighted and heated in a platinum pan under air (60 mL.min^−1^) at a heating rate of 10 °C.min^−1^.

#### 3.4.6. Cobb Test for Liquid Water Absorption

The liquid water absorption was measured according to the TAPPI standard method T 441 om-98. Briefly, a ring with a test area of 50 cm^2^ was placed on the coated paper, and 50 mL of distilled water was added. After 60 s of contact, water was poured out, and excess of water was removed from surface using blotting paper and a heavy weight roll. The water absorption was calculated from the weight increase of paper over test area surface. Cobb values were obtained from five repetitions.

#### 3.4.7. Water Contact Angle

Water contact angle measurements were carried out using a goniometer Krüss DSA 100 (Hamburg, Germany). A drop (3 µL) of distilled water was placed at the surface of coated papers, and the measurement was taken after five seconds. The resulting contact angle was obtained from ten measurements.

PCC particles were spread on a double pressure sensitive adhesive tape after sticking on one side on a glass slide. Powder excess was removed by air flushing.

#### 3.4.8. Penetration Dynamic Analysis (PDA)

Water absorptiveness in materials was followed using Emtec PDA C02 apparatus (Emtec Electronic, Liepzig, Germany). A paper sample was brought into contact with water in a measuring cell. During water contact, the sample was crossed with high frequency low-energy ultrasonic signals. These signals were received by a sensor before they were processed in the device and analyzed by a computer. PDA curves were plotted as the percentage of transmission of ultrasonic signals vs. time. Transmission at t = 0 s was set at 100%. The time corresponding to maximum of each curve was defined as the wetting time of the material. The slope of the curve after the maximum was proportional to the speed of the water penetration. Each curve was the average of three experiments.

#### 3.4.9. Water Vapor Permeability

Water vapor permeability experiments were carried out according to the method NF ISO 2528 (1995). An aluminum cup with 20 g of anhydrous CaCl_2_ was sealed using paper material with melted paraffin wax. Cups were then placed in a climatic chamber at 25 °C, 50% RH. The water vapor transmission rate (*WVTR*) was calculated from the weight increase of the cup over time using the following equation:(1)$$WVTR=\frac{\Delta m.24}{\Delta t.S}\;in\;g.m^{-2}.d^{-1}$$
where Δm is the weight (g) of water vapor passing through a specific area *S* (50 cm²) of the material during a time, Δt (h). Plotted-linear regression was used to calculate the slope of a fitted straight line. A correction of the paper weight gain or loss was conducted in parallel with control cup without desiccant agent.

#### 3.4.10. Scanning Electron Microscopy (SEM) Views

Surface and cross-section microstructure analyses were carried out using SEM technique with a tungsten source in a Quanta 200 device (FEI Company, Hillsboro, OR, USA). Samples were observed using low vacuum 50 Pa, large field detector, 4–7 kV and working distance 9–11 mm.

#### 3.4.11. Low Field Relaxometry

NMR measurements were carried out on a Bruker Minispec mq (mq20, Bruker, Billerica, MA, USA) 60 operating at 60 MHz proton resonance frequency corresponding to a magnetic field of 1.41 T. π/2 pulse length is 2.8 µs, and the typical value of dead time is 7 µs. The temperature of the magnet, the probe temperature and samples (around 20 mg) were thermalized to 37 °C. Before each measurement, a duration of 30 min was allowed for temperature equilibration of the samples.

Carr–Purcell–Meiboom–Gill sequence (CPMG) was used for the T_2_ study. Spectra were acquired setting 600 echoes, with an accumulation of 1024 scans spaced out by a recycle delay of 3 s. Echo spacing of 40 µs. Signal treatment was operated using Minispec software giving discrete values of relaxation time and corresponding intensity.

#### 3.4.12. Mechanical Tests

Mechanical properties were performed on an MTS QTest 25 Elite controller (MTS System Corporation, MN, USA). Conditioned papers were cut in strips (15 mm width × 100 mm long). The initial grip separation was set at 50 mm, and the crosshead speed at 10 mm.min^−1^. All tests were conducted in the machine direction. Five replicates of each material were used for analyses.

#### 3.4.13. Microorganisms and Culture Conditions

*C. globosum*, ATCC 6205 (Institut Pasteur, Paris, France), was used for the antifungal test. The strain was grown on potato–dextrose–agar Petri dish and incubated at 25 °C, 70% RH until sporulation. Then, the spores were dispersed in physiological water with 0.1% Tween 80. After spore numeration using a Malassez cell, the concentration was adjusted to 2000 spores.mL^−1^ using physiological water. To evaluate the antifungal activity, discs of coated paper (diameter: 50 mm) were deposited on potato–dextrose–agar medium at the center of Petri dishes. The 10 µL of fungal dispersion was deposited at the center of papers before incubation at 25 °C, 70% RH. The fungal growth diameter was measured every day on three replicates.

#### 3.4.14. Statistical Treatment

All experiments were repeated at least three times. Data were treated using the Student’s t-test (*p* < 0.05).

## 4. Conclusions

The modification of calcium carbonate particles by stearic acid and their incorporation in a chitosan film-forming solution led to an increase in the surface hydrophobicity mainly through the control of roughness. Selected formulations containing 20 and 50 wt.% of modified PCC were coated on paper. The modified particles counterbalanced the chitosan hydrophilicity coating and contributed to controlling the coatings’ penetration in the paper bulk. The highest roughness was obtained with the 50 wt.% of modified PCC formulation, increasing the liquid water resistance and wetting time. This coating has also exhibited better mechanical properties.

The antifungal activity of the film-forming solution against *C. globosum* was evaluated and showed a possible synergistic effect between chitosan and bicarbonate ions from the mineral particles. The film-forming solution with 20 wt.% of modified PCC provided the highest lag phase delay. However, coated paper showed a limited antifungal bioactivity due to the low dry matter coated. To combine the water resistance of the paper coated with 50 wt.% of modified PCC formulation and good antifungal bioactivity, the film-forming solution has to be enhanced. As already reported in the literature, chitosan bioactivity could be enhanced by quaternization [49] or by using a low molar mass such as oligochitosan [50].

## Figures and Tables

**Figure 1 molecules-27-08886-f001:**
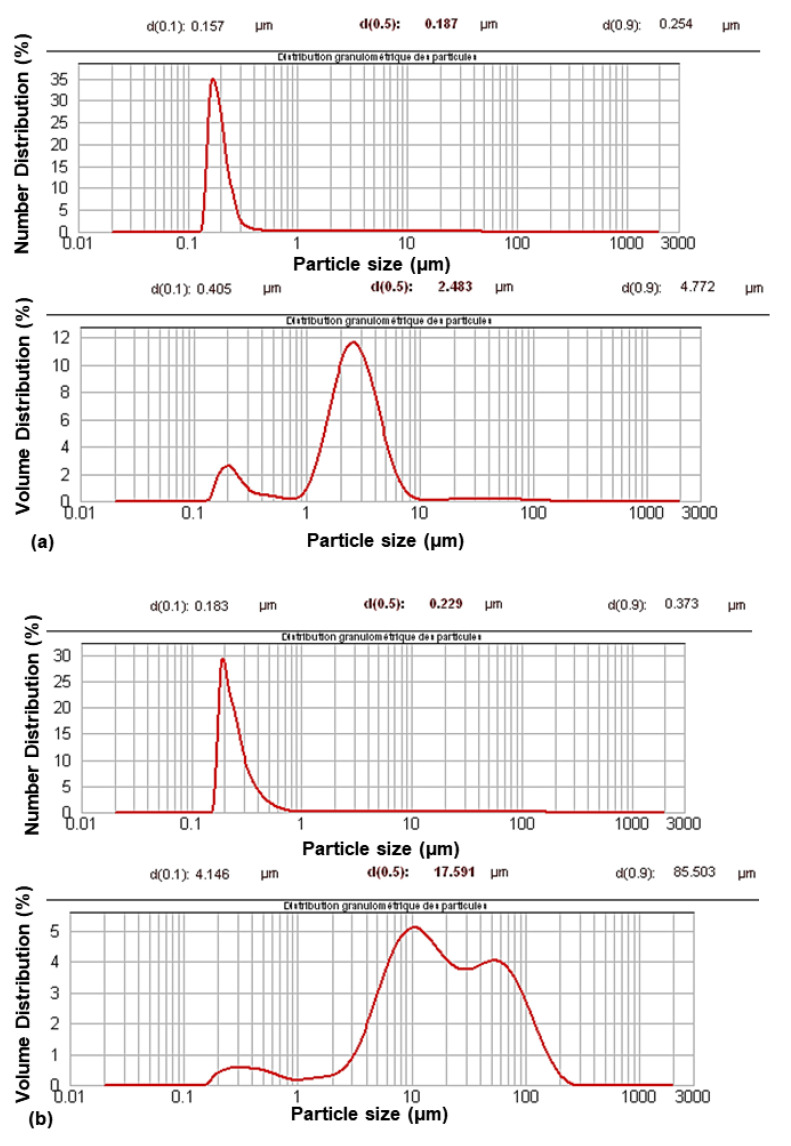
Granulometric curves of PCC before (**a**) and after modification (**b**).

**Figure 2 molecules-27-08886-f002:**
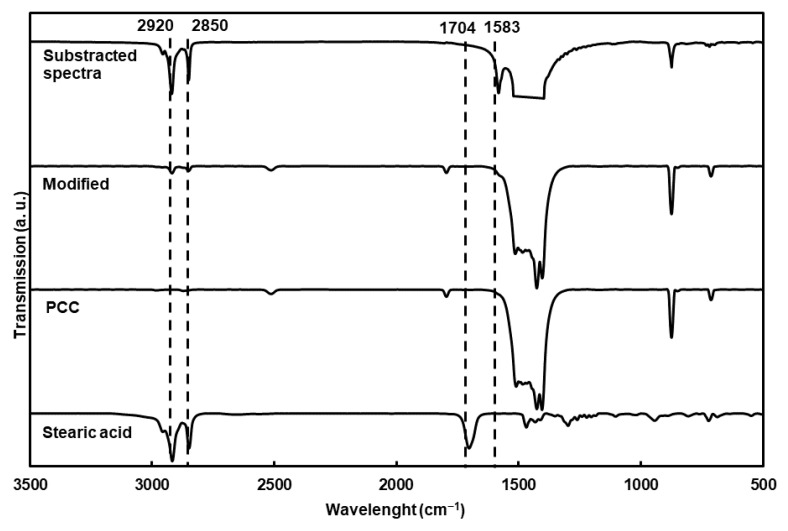
FTIR transmission spectra of modified PCC, stearic acid and PCC. The spectrum of modified PCC spectrum was substracted from PCC spectrum, but the band between 1400 and 1500 cm^−1^ was cut for clarity.

**Figure 3 molecules-27-08886-f003:**
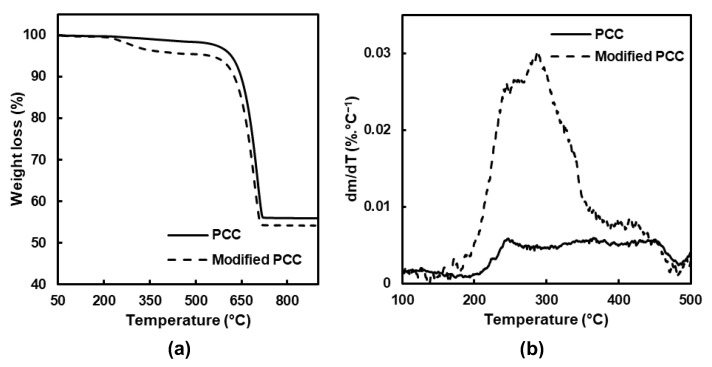
(**a**) TGA and (**b**) dTGA of PCC before and after modification.

**Figure 4 molecules-27-08886-f004:**
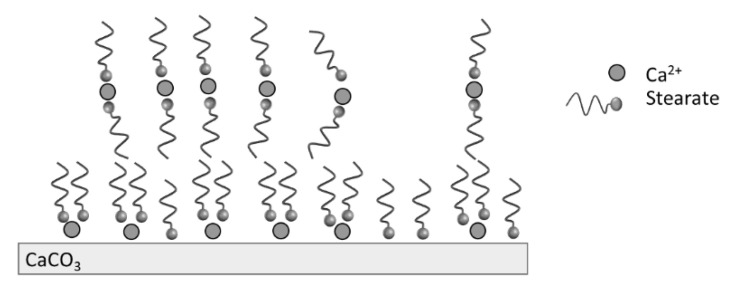
Illustration of stearate adsorption at the calcite surface.

**Figure 5 molecules-27-08886-f005:**
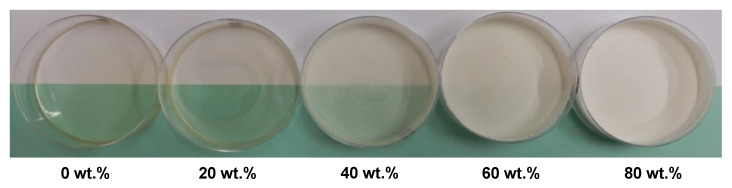
Visual aspect of chitosan films prepared with different amounts (wt.%) of modified PCC. Chitosan was first solubilized (1.0% *w*/*v*) in lactic acid solution (0.5 vol.%) before incorporation of modified PCC.

**Figure 6 molecules-27-08886-f006:**
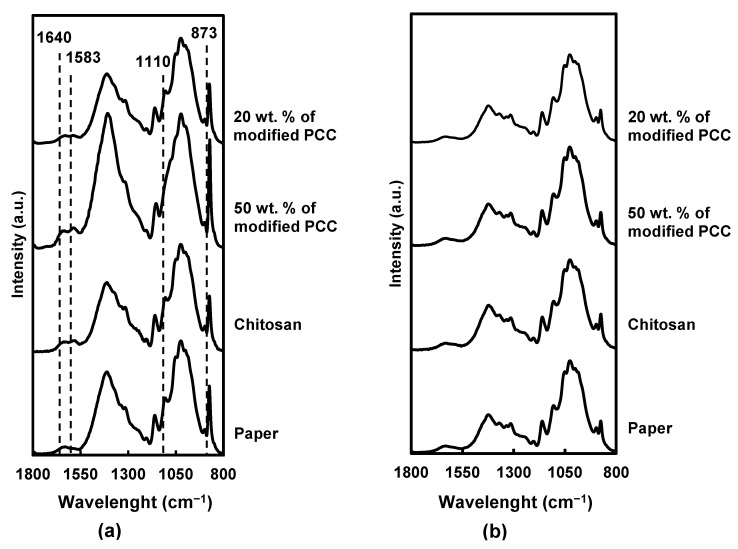
FTIR spectra for paper and coated paper. (**a**) Coated side, (**b**) uncoated side.

**Figure 7 molecules-27-08886-f007:**
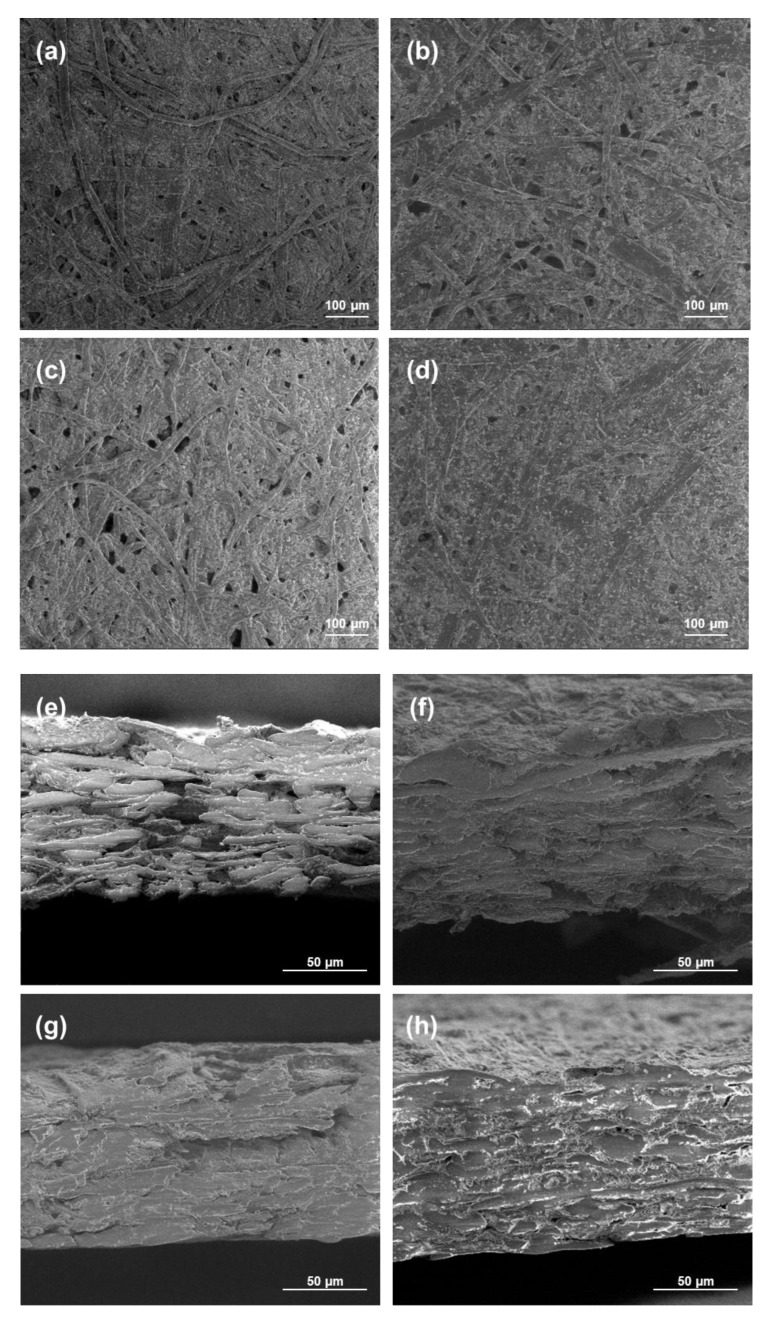
SEM micrographs of surface and cross-section of paper samples. For cross-section images, coated sides were on the top. (**a**,**e**) Uncoated paper; (**b**,**f**) 20 wt.% of modified PCC; (**c**,**g**) chitosan coating; (**d**,**h**) 50 wt.% of modified PCC.

**Figure 8 molecules-27-08886-f008:**
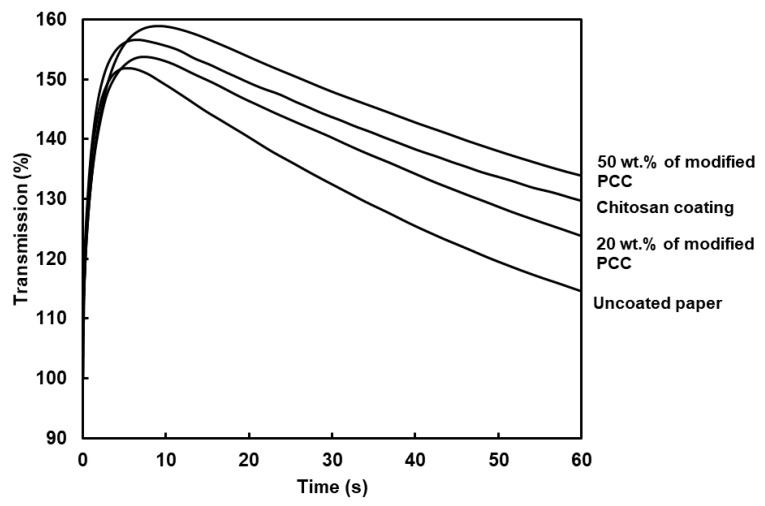
Curves of dynamic liquid water penetration for paper substrate and coated papers. Coatings were made from film-forming solutions containing only chitosan or chitosan and modified PCC at 20 and 50 wt.%.

**Table 1 molecules-27-08886-t001:** Weight loss of PCC (wt.%) before and after modification between 180 and 480 °C.

	180–380 °C	380–480 °C
PCC	0.85	0.50
Modified PCC	3.54	0.62

**Table 2 molecules-27-08886-t002:** Water contact angle of composite films with modified PCC.

PCC Modified (wt.%)	Water Contact Angle (°)
0	56 ± 2
20	77 ± 5
40	79 ± 3
60	68 ± 5
80	Direct absorption

**Table 3 molecules-27-08886-t003:** Liquid water resistance of coatings evaluated by various physicochemical characterizations.

	Thickness (µm)	Grammage of Coatings (g.m^−2^)	Water Contact Angle (°)	Bendtsen Rugosity (mL.min^−1^)	W	Wetting Time (s)	Cobb_60_ (g .m^−^²)
20 wt.% of modified PCC	124 ± 2	0.6 ± 0.3	98 ± 3	920 ± 30	62.19	7.4	17.3 ± 0.8
50 wt.% of modified PCC	125 ± 2	0.7 ± 0.3	110 ± 5	950 ± 50	83.29	9.1	17.0 ± 0.4
Chitosan coating	125 ± 3	0.7 ± 0.4	82 ± 3	950 ± 20	52.69	6.5	17.0 ± 0.6
Paper	127 ± 3	/	110 ± 4	900 ± 40	44.09	5.3	20.0 ± 0.1

**Table 4 molecules-27-08886-t004:** Water vapor transmission rate of paper substrate and coated papers (23 °C, 50% RH).

	Water Vapor Transmission Rate (g.(m².d)^−1^)
20 wt.% of modified PCC	724 ± 3
50 wt.% of modified PCC	735 ± 7
Chitosan coating	669 ± 14
Paper	814 ± 7

**Table 5 molecules-27-08886-t005:** *T*_2_ relaxation spin–spin for paper, coatings and chitosan film. Intensities of signals are weighted by sample water content. *W*_a_ (respectively, W_b_) was associated with the ratio intensity of the signal corresponding to *T*_2*a*_ (respectively, *T*_2*b*_).

	Water Content (%)	*T*_2*a*_ (ms)	W_a_ (%)	*T*_2*b*_ (ms)	W_b_ (%)
20 wt.% of modified PCC	5.0	0.92 ± 0.01	92.6	12.90 ± 0.60	7.4
50 wt.% of modified PCC	5.2	0.76 ± 0.01	92.9	13.30 ± 0.60	7.1
Chitosan coating	7.2	0.77 ± 0.02	90.7	13.00 ± 0.50	9.3
Chitosan film	22.5	0.12 ± 0.01	95.4	11.80 ± 0.40	4.6
Paper	4.3	1.00 ± 0.01	92.3	13.20 ± 0.50	7.7

**Table 6 molecules-27-08886-t006:** Mechanical properties of paper and coated paper.

	Young’s Modulus (GPa)	Elongation at Break (%)	Tensile Strength (Mpa)
20 wt.% of modified PCC	4.6 ± 0.2	3.0 ± 0.4	69.7 ± 1.8
50 wt.% of modified PCC	4.5 ± 0.3	3.0 ± 0.1	70.7 ± 4.3
Chitosan coating	4.4 ± 0.4	3.0 ± 0.2	60.1 ± 3.6
Paper	3.8 ± 0.3	2.8 ± 0.2	39.0 ± 3.4

**Table 7 molecules-27-08886-t007:** Antifungal activity of film-forming solutions on *C. globosum* (20 spores inoculated) at days 3 and 11. Inhibition percentages were given as control reference.

		Lag Phase (d)	Inhibition at Day 3 (%)	Growth Rate (mm.d^−1^)	Inhibition at Day 11 * (%)
Chitosan	20 wt.% of modified PCC	9	100	22.0 ± 1.9	40
	50 wt.% of modified PCC	2	6	20.5 ± 1.3	0
	Control (pH 4.2)	4	94	17.0 ± 0.4	0
	Control (pH 6.2)	4	100	16.3 ± 1.4	0
HPMC	20 wt.% of modified PCC	2	−24	18.0 ± 1.3	0
	50 wt.% of modified PCC	2	−16	18.5 ± 0.4	0
	Control (pH 4.2)	2	0	17.1 ± 0.4	0
	Control	2	/	18.4 ± 0.4	/

* At day 14, all Petri dishes showed no inhibition.

**Table 8 molecules-27-08886-t008:** Antifungal activity of coated paper on *C. globosum* (20 spores inoculated) at days 3 and 11. Inhibition percentages were given as control reference.

	Lag Phase (d)	Inhibition at Day 3 (%)	Growth Rate (mm.d^−1^)	Inhibition at Day 11 * (%)
20 wt.% of modified PCC (one layer)	2	19	18.4 ± 1.7	0
20 wt.% of modified PCC (two layers)	2	44	10.0 ± 0.8	0
50 wt.% of modified PCC (one layer)	2	32	13.0 ± 0.8	0
Chitosan coating	2	11	16.7 ± 1.2	0
Paper	2	/	23.5 ± 2.3	/

* At day 14, all Petri dishes showed no inhibition.

## Data Availability

Not applicable.

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
