# Peer review of "Functional Chitosan–Calcium Carbonate Coatings for Enhancing Water and Fungal Resistance of Paper Materials"

_molecules, 2022, doi:10.3390/molecules27248886_

Round 1
Reviewer 1 Report
The manuscript “ Functional chitosan-calcium carbonate coatings for enhancing water and fungal resistance of paper materials” was reviewed for consideration in Molecules. Following points should be considered
Keywords: The keywords should be chosen in such a way, that it should not be matched with the titled.
Fig 4: The figure is not so clear, please make it to more visible and clearer.
Line 209: Please remove Chinese words.
Table 7: It should be worth mentioning to include days as variable too.
Section 3.4.1 to 3.4.13, no reference was added in any analytical procedure, please explain?
Comment: Line 428: Data are mean values given with ….. sentence is not grammatically correct, please correct
Author Response
Reviewer #1:
First, the authors thank the reviewer 1 for his/her constructive remarks
All the suggestions and comments were taken into consideration.
More specifically:
- Keywords
The authors have modified keywords as suggested the reviewer.
- 4.
As suggested, the authors improved the clarity of the Fig. 4.
- Line 209
The authors remove non English characters.
- Table 7 and 8
The authors have added days as variable.
- Reference in analytical procedure
The modification of PCC was described in the literature and cited in the References section. Other analytical procedures or analyses were described in the Materials and Methods section.
- Line 428
The sentence was corrected as suggested
Reviewer 2 Report
Hydrophobicity is generated when there is a combination of micro and nanostructures that is called hierarchical structure.
line 33: micro and nanostructured
line 209: remove non English characters
In the contribution section there are no names
Author Response
The experimental part was much more detailed as suggested.
Reviewer #2:
First, the authors thank the reviewer 2 for his/her comments and remarks to improve the manuscript
All the suggested modifications and all the comments were taken into consideration
- Line 33
The sentence was modified as suggested.
- Line 209
The authors remove non English characters.
- Contribution section
The authors have added names in this section as required.
Reviewer 3 Report
l They did not give the essential information of the PCC they used; their mean size, size distribution, and morphology (micrographs). Their experimental results should be changed every time when they use different PCC.
l Authors claims that the rugosity of the PCC caused the hydrophobicity. However, it was difficult to differentiate the effect of hydrophobic polymers attached to the PCC and the rugosity of the PCC for developing hydrophobicity of the film and the coating on the paper materials. More modified PCC means more hydrophobic polymers attached to it.
l Table 3 did not support what authors want to say. More rugosity resulted in more absorption of water (higher Cobb60. Lines 197-203).
l Coating with 0, 20 and 50% PCC containing chitosan actually modified very insignificantly within experimental error, the hydrophobicity and antibacterial properties of the paper, but developed tensile strength more than 50%. If the effect of the modified PCC was insignificant in developing hydrophobicity, it should be better to suggest the chitosan coating only.
l In Abstract and Conclusion, they claimed that the surface hydrophobicity of paper was increased by the increase of rugosity or surface roughness provided by the PCC. But I cannot find no direct proof.
Author Response
The experimental part was much more detailed as suggested.
Reviewer #3:
We thank the reviewer 3 for his/her constructive remarks.
All the suggestions and comments were taken into consideration
More specifically, the particle size of the PCCs before/after modification was added and the Table 3 was modified as suggested.
Some specific comments are listed below:
- Informations about PCC used
The authors have added the granulometric curves of the PCC before and after modification.
- Association of rugosity and hydrophobicity
See next comment.
- Table 3
Table 3 was modified as follows:
A column was unfortunately missing in the original Table 3, which led to a shift. Moreover, some data were added from PDA analyses (Table 3, column W).
Surface wettability was given by factor “W”: the higher the value, the lower the surface wettability. Data showed that surface wettability was not only dependent to surface hydrophobicity given by WCA measurements. The dynamic water penetration was related to the evacuation of air layer at the surface of the paper. Combination of higher rugosity and higher WCA led to a lower surface wettability.
- Mechanical properties developed by chitosan
Tensile strength of coated paper was higher using modified PCC than chitosan alone.
Round 2
Reviewer 1 Report
The revised version was reviewed, and all the comments/ suggestion has been incorporated. The manuscript is ready for publication.